# Viral Biomarker Detection and Validation Using MALDI Mass Spectrometry Imaging (MSI)

**DOI:** 10.3390/proteomes10030033

**Published:** 2022-09-13

**Authors:** Matthew B. O’Rourke, Ben R. Roediger, Christopher J. Jolly, Ben Crossett, Matthew P. Padula, Phillip M. Hansbro

**Affiliations:** 1Centenary UTS Centre for Inflammation, Faculty of Science, School of Life Sciences, Centenary Institute and University of Technology Sydney, Sydney, NSW 2007, Australia; 2Centenary Institute, Faculty of Medicine and Health, The University of Sydney, Camperdown, NSW 2006, Australia; 3Novartis Institutes for Biomedical Research-Autoimmunity, Transplantation and Inflammation, 9656 Basel, Switzerland; 4Lowy Cancer Research Centre, School of Medical Sciences, UNSW Sydney, Sydney, NSW 2007, Australia; 5Sydney Mass Spectrometry, University of Sydney, Camperdown, NSW 2006, Australia; 6Metabolomics and Lipidomics Core Facility, Faculty of Science, School of Life Sciences and Proteomics, University of Technology Sydney, Ultimo, NSW 2007, Australia

**Keywords:** mass spectrometry imaging, peptide identification, MALDI

## Abstract

(1) Background: MALDI imaging is a technique that still largely depends on time of flight (TOF)-based instrument such as the Bruker UltrafleXtreme. While capable of performing targeted MS/MS, these instruments are unable to perform fragmentation while imaging a tissue section necessitating the reliance of MS1 values for peptide level identifications. With this premise in mind, we have developed a hybrid bioinformatic/image-based method for the identification and validation of viral biomarkers. (2) Methods: Formalin-Fixed Paraffin-Embedded (FFPE) mouse samples were sectioned, mounted and prepared for mass spectrometry imaging using our well-established methods. Peptide identification was achieved by first extracting confident images corresponding to theoretical viral peptides. Next, those masses were used to perform a Peptide Mmass Fingerprint (PMF) searched against known viral FASTA sequences against a background mouse FASTA database. Finally, a correlational analysis was performed with imaging data to confirm pixel-by-pixel colocalization and intensity of viral peptides. (3) Results: 14 viral peptides were successfully identified with significant PMF Scores and a correlational result of >0.79 confirming the presence of the virus and distinguishing it from the background mouse proteins. (4) Conclusions: this novel approach leverages the power of mass spectrometry imaging and provides confident identifications for viral proteins without requiring MS/MS using simple MALDI Time Of Flight/Time Of Flight (TOF/TOF) instrumentation.

## 1. Introduction

Matrix-Assisted Laser Desorption-Ionisation Mass Spectrometry Imaging (MALDI-MSI) combines visual representation and spatial mapping of molecules by histology together with sequencing and discovery-based identification of molecules using modern mass spectrometry “omics” style techniques [1]. The nature of such a hybrid analytical modality has often made the data challenging to interpret, since it requires both a qualitative interpretation of its images and further bioinformatic interpretation of the raw mass spectrometric data [2]. 

The most common form of MALDI-based imaging platforms is the simple MALDI TOF/TOF instrument. These workhorse mass spectrometers are relatively easy to operate; however, for their reliability and ease of use, there is generally a sacrifice in mass resolving power. Typical Orbitrap, Quadrupole Time Of Flight (QTOF) or Fourier Transform Ion Cyclotron Resonance (FTICR) systems can offer anywhere between 50,000 to >1,000,000 mass resolution [3,4], which is enough for molecular identification from single MS scans, whereas typical TOF/TOF systems generally achieve between 5000–40,000 resolution [5] which is not high enough to determine molecular identity by mass alone. The addition of ion fragmentation and tandem mass spectrometry (MS/MS) means that for a standard preparation, a MALDI TOF/TOF can perform peptide sequencing, where the issues of mass resolution are less important. However, when a MALDI TOF/TOF is used for imaging, the workflow relies on a single MS scan where only the parent mass is detected, with no fragmentation performed and identification reliant on the highest possible mass resolution of the parent molecular mass [6].

Approaches that rely on identification by measurement of intact molecules by mass spectrometry (MS1 values) are well published [7]. However, in recent years, the need for very high mass resolution has been clearly identified as critical to ensuring that the matching of those values with in-silico or theoretically calculated peptide values is accurate. This, in turn, has created a new requirement that any MSI experiment performed using a simple MALDI TOF/TOF system requires orthogonal validation with either complementary Liquid Chromatography tandem Mass Spectrometry (LC-MS/MS) workflows or another imaging modality such as immunofluorescence [8]. The ongoing problem is matching the peptides detected via LC-MS to MS1 values from the MALDI instrument or finding an antibody that can be purchased or created that is specific for the molecules of interest [9]. This is further complicated when the sequence of the peptide is unknown due to a lack of fragmentation data and hence, raising an antibody becomes extremely difficult, not to mention the costs, labour and time required for antibody development and validation. 

Another approach employs tissue fragmentation, whereby once MS1 scans are acquired, those same masses are then manually acquired from the same tissue section with fragmentation enabled [10]. Although the spatial information is lost when this is performed, the laser path that ionises the sample can be confined to a small area of tissue, thereby ensuring that the peptides that are fragmented originate from the region where they were first detected. This approach has been reported to be relatively accurate [11]; however, the need to manually acquire the data and then subsequently search it is time consuming and labour intensive. An added frustration with this approach is that, since ions are generated in a plume, the ability to only analyse a single targeted molecule is limited. This is especially true in lipid imaging, where a target mass ±0.1 m/z could result in the co-fragmentation of several structurally distinct lipid species [12]. This then produces complex fragmentation spectra that is difficult to deconvolute in order to assign the applicable fragments to their parent molecules. 

Our laboratory has struggled with these difficulties since our first peptide imaging study in 2015 [13], and since then there have been few significant advances in MALDI TOF/TOF technology. Furthermore, a trend has emerged where, rather than having a dedicated instrument for MSI, ion sources and attachments can be purchased that modify existing QTOF or Orbitrap instruments [6,14]. While this is certainly an advance, there is still a significant number of simple TOF/TOF instruments in use. With all of this in mind, our team has developed a novel approach for confirming and validating peptides that are detected using a standard TOF/TOF MSI experiment. This workflow involves integrating visual image analysis and bioinformatic determination of protein identifications along with our previously published internal controls to ensure correct sample preparation.

Most recently, we have applied this workflow to characterising the presence of a viral infection in mouse kidneys. Parvovirus are two-gene viruses, comprising NS1 and VP1, with this particular virus showing a high degree of divergence from other well-known mouse parvoviruses [15]. This novel parvovirus was first described in 2018 by Roediger et al. [16]. MSI was performed on serial tissue sections derived from the same samples that were used in this previous publication. The use of these samples and parallel analyses gave us a unique opportunity to test and validate the efficacy of this new MSI workflow, as described below.

## 2. Materials and Methods

All mouse samples were initially generated for a previous study and as such no new mice were required for this analysis. The details of husbandry and maintenance were as previously described [16]. Additionally, all infection models, pathological investigations and verifications by PCR and genome sequencing have also been validated and published previously [16,17].

### 2.1. Tissue Preparation for Imaging

Tissue samples were prepared and analysed according to a well-established protocol from O’Rourke et al. (2018) [18]. Paraffin-embedded sections were cut at 10 µm and float mounted onto liquid nitrocellulose-coated indium tin oxide slides and allowed to dry in a vacuum desiccator for 24 h. To prepare tissue prior to tryptic digestion, mounted samples were immersed in fresh xylene for 2 min to remove paraffin, then de-lipidated and de-salted in a 6-stage wash protocol of 70% EtOH, 100% EtOH, Carnoy’s Fluid, 100% EtOH, Water and 100% EtOH. Each stage lasted for 30 s except for Carnoy’s fluid, which lasted 2 min. Carnoy’s fluid was made by mixing 100% Ethanol, Chloroform and glacial Acetic Acid in a 6:3:1 ratio. Samples were then subjected to methylene hydrolysis to remove formaldehyde crosslinks by immersing the samples in 20 mmol Tris-HCl (pH 8.8) in a closed slide box. The box was then placed in 500 mL of water and heated for 15 min at 120 °C in a commercially available pressure cooker(Kambrook KPR620BSS, Sydney, Australia) at an operating pressure of 70 Kpa. Pressure-cooked samples were dried in ambient conditions and proteolytically cleaved by pipetting 10 µL of 1 mg/mL (aq) trypsin onto the centre of the tissue sections, and using the longitudinal side of the pipette tip and surface tension to drag the trypsin solution droplet to cover the whole of the sections. Slides were allowed to dry at ambient temperature before being mounted into the upper part of a vapour chamber sealed with parafilm, as previously described [13]. Finally, 650 µL of a 50:50 mix of 50 mmol ammonium bicarbonate/100% Acetonitrile was pipetted evenly onto the centre paper tabs, the chamber was then assembled, sealed with parafilm and left to incubate at 37 °C overnight (12–16 h).

### 2.2. Sublimation

Following digestion, the samples were coated in α-Cyano-4-hydroxycinnamic acid matrix for the ionisation of peptides. Slides were weighed and then affixed to the inside of the cooling finger of the sublimator (ChemGlass life sciences, Vineland, NJ, USA). Matrix (500 mg) was placed in the bottom petri dish creating an even coating and the chamber was sealed and the vacuum engaged. After 5 min, the cooling finger of the sublimator was packed with ice and 50 mL of water was added. The chamber was allowed to settle for a further 5 min prior to sublimation, which was performed for 45 min in a 260 °C sand bath. This achieves an ideal coating of 0.2 mg/cm^2^. Once sublimation was complete, the chamber was vented and immediately disassembled to prevent condensation formation. The sample was removed and re-weighed to ensure correct coverage and then re-placed into the vapour chamber with 650 µL of 50%ACN: 0.1%TFA pipetted evenly onto the paper tab. The chamber was assembled (without the parafilm) and the matrix left to re-crystallise with the sample at 37 °C for 1 h. 

### 2.3. MALDI Instrumentation

Once re-crystallised, samples were mounted into a slide adaptor bracket and were imaged in an UltraFlextreme MALDI TOF/TOF (Bruker Daltonics Breman Germany) with the following settings: reflector positive mode, laser power—65%, laser attenuation—30%, detector gain—27×, mass range—750–3500 m/z, sample rate/Digitizer—1.25 GS/s, realtime smoothing—Off, smartbeam parameter set—2_small, frequency—1000 Hz, laser Shots—500 and raster width 50 µm. Prior to imaging, a co-mounted serial section was used for on tissue calibration using the same settings: 1 µL of Standard Peptide Mix 2 (Sciex, Framingham, MA, USA) was pipetted onto the tissue surface and allowed to dry. Calibration was performed until a <50 ppm accuracy was achieved. 

### 2.4. Data Analysis MSI

After acquisition, the data was processed, as described previously [12], using Scills Lab 2014 b, with the added step of importing all experimental data and then normalizing to the total ion count of the global spectrum. Briefly, each segmented image was then validated manually by inspecting the associated peak in the global mass spectra to ensure that the entirety of the peak width had been integrated. Images were also inspected to ensure that the detected peak was genuine and not an artifact or background noise, as described previously [19,20]. This involved imaging an area outside the borders of the tissue and checking the images detected by segmentation to ensure that the ion signals identified were confined to the tissue borders and did not extend beyond. Ion data that appears outside the tissue is not of biological origin and is the result of matrix ion clusters or electrical noise. The resulting validated images were then exported as grey scale. TIF files are imported into Image J [21], where pixel-by-pixel correlations were performed between selected images based on signal intensity. Since each image only contained a single m/z value, there was no expectation of interference from other signals.

### 2.5. Data Analysis Peptide Mass Fingerprinting

Once image analysis was complete, the m/z values for each validated image were exported and then compared to the m/z values of a theoretical digest of the target proteins: NS1 and VP1. The suspected viral peptide m/z values were then searched with MASCOT [22] against the *Mus musculus* proteome spiked with amino acid sequences of both VP1 and NS1. A matched decoy list was also included as a negative control. The resulting searches were then performed with the following settings: Enzyme: trypsin, missed cleavages: 2, Variable modifications: Oxidation (M), Peptide tolerance 1.2 Da, monoisotopic masses only, mass value [M + H]^+^.

## 3. Results

### 3.1. Peptide Mass Fingerprinting

A theoretical in silico digest of both target viral proteins was performed using the ExPasy peptide cutter. The subsequent list of tryptic peptides was then compared to the mass list generated from the global spectrum of the MSI analysis run (Figure 1). This allowed for rapid targeted screening of images associated with these potential viral peptides. Peptide mass fingerprinting (PMF) was then used to screen detect MS1 masses and revealed the detection of 14 peptides from target viral proteins. This included confident identifications of 10 peptides from NS-1 influenza protein (NS1) and four from the major Capsid Protein VP-1 (VP-1) (Table 1). A non-viral mouse protein (TIFIIA2) was also identified using an untargeted search of all detected masses not included in the targeted viral analysis against the mouse FASTA database and was subsequently used as a negative control in the MSI correlational analysis (Table 1). 

### 3.2. Image Correlation Analysis

MSI image analysis was performed on the same peptides detected via PMF. Following the extraction of the matched images for each peptide, a Pearson’s correlation was performed to compare the intensity of the isolated peptide at each X/Y coordinate. Across each image, a total of ~35,000 data points were compared, which yielded an R^2^ range of between 0.79–0.92 for NS-1 peptides (Figure 2A) and 0.84 for the two detected VP-1 peptides (Figure 2B). A further comparison was performed between two peptides from both NS1 and VP1 (Figure 2C), yielding an R^2^ of 0.93 confirming that both viral proteins were present at the same location throughout the image. The same was performed for two peptides from the host transcription factor TFII2A (Figure 2D), returning an R^2^ of 0.92 confirming that both of its peptides also colocalised. A final correlation analysis was performed between viral NS1 and the TFIIA2 mouse proteins, as a negative control. This yielded an R^2^ of only 0.32 (Figure 2E). This strongly suggests that the co-localised peptides for both NS1 and VP1 did in fact belong to the viral proteome, since they did not co-localise with endogenous mouse proteins that correlate based on the gross macrostructure of the tissue, as the viral infection is locationally nonspecific. The correlational analysis of all peptides can be found in Appendix A.

## 4. Discussion

The aim of this study was to spatially map the presence of a virus that causes chronic nephropathy and eventually death in immunodeficient mice. Our previous study successfully characterised a novel parvovirus that specifically infects murine kidneys at a sufficient viral load that viral peptides could be detected in moribund mice by conventional MS/MS [16]. This led us to assess whether the presence of the virus could be detected via MALDI MSI and subsequentially spatially mapped.

The first consideration with any MSI workflow is the application of the correct sample preparation, and the possible downstream effect that this chemical processing will have on the resultant data. The biggest source of potential chemical interference is during the methylene hydrolysis of formalin fixation, which can often leave peptides still crosslinked to other peptides with methylene bridges, or leave residual chemical groups on primary amines, which adjusts the mass [23,24]. In MALDI MSI, the identification of peptides is determined by the mass of the intact peptide and therefore any mass that still contain residual methylene crosslinks or partially bound formaldehyde will not match to the in silico digest used as a reference list. For our study, this did not prove fatal, since we still achieved a large amount of confidentially identified peptides; though this did only represent a small fraction of the complex peptide spectra generated. In our opinion, more work is needed to better explore this type of data, since there were likely additional viral peptides that were not detected due to the presence of variable post translational modifications caused by formaldehyde fixation. 

The importance of reliably identifying peptides is a critical aspect of any MALDI MSI workflow. Methods using orthogonal validation such as LCMS still have the underlying problem of attempting to match complex peptide lists of multiply charged peptides to singly charged m/z lists generated by TOF/TOF instruments. While ultra-high resolution FTICR or MALDI sources coupled to QTOF instruments overcome this issue, the expense and complexity of these systems make them prohibitive to the majority of users and institutions. Here, we utilise a hybrid identification workflow to achieve reliable results together with the integration of inbuild validation steps throughout the workflow.

One particular aspect of validating MSI data is the difficulty in integrating hybrid techniques of analysis [2]. Primarily, MALDI MSI is a mass spectrometry technique; however, the resulting data is not a traditional list of m/z values, retention times and MS2 fragments. Instead, the result of a complete MSI workflow is a collection of m/z values that are contained with specific XY coordinates, resulting in visible pictures at specific m/z values. The lack of MS2 fragments and accompanying spatial information make many analysis pipelines inappropriate; however, the ability to generate images allows other methods of analysis that can self-validate the data without the use of any orthogonal workflows. 

The basis of this approach is centered around analysing MS data using one method and then validating what has been identified using the accompanying images. This is based on the hypothesis that peptides from the same protein should occur in the same physical location and at a consistent relative abundance across each XY coordinate in an image. 

Using the example of VP1, we found that the m/z values 1524.7805, 919.4996, 838.5145 and 780.4111 were all present in the global spectrum. To determine if these were actually viral and not mouse proteins, and in the absence of any MS2 fragments, we performed a PMF. To avoid biasing the search algorithm, we also included a list of masses that were also observed in the global spectrum but were not from either viral protein. The resulting expectation score was statistically significant (>25 LogP), indicating that the identification was correct. 

One consideration with the detection of peptides using MSI is that a traditional PMF style search query does not normally account for the presence of post translationally modified proteins or proteoform variations of the base sequence of the targeted viral proteins.

In our workflow, the peptides are identified by mapping them back to a theoretical digest of the targeted viral proteins. The possibility of modified peptides within those proteins being present is likely. However, due to the efficiency of ionisation of modified peptides and their relative low abundance, it is unlikely that they will be detected in an MSI-based workflow. Further to this, limitations with mass resolution achievable in a TOF/TOF instrument makes assigning modified peptides to viral proteoforms unreliable. More work is needed in this area in order to confidently search and identify proteoforms using MALDI TOF/TOF MSI.

In workflows such as this, and indeed in the majority of proteomics workflows, it is desirable to have some form of validation. Orthogonal techniques to MSI can often introduce unnecessary complexity and therefore we used the generated images to validate our findings. To do this, we treated imaging data as if it had originated from some form of microscopy and applied a well-known analysis or a correlation of each pixel from each m/z image from the suspected viral peptides. 

The utility of this image analysis was enhanced by the fact that all images were generated from a single acquisition. Since the XY co-ordinates and number of pixels does not change between images, and it is not necessary to designate regions of interest to account for varying numbers of pixels in each image, the analysis can perform a direct comparison of the intensity of each individual pixel across each image and compute a reliable R^2^ co-efficient. The result of this correlational analysis demonstrated clearly that the peptides that were considered to belong to the viral proteins co-occurred with similar intensity with a correlation R^2^ value of between 0.79 and 0.92 for all peptides and did not co-occur with the negative control peptides from TFIIA, indicating that they followed the pattern of viral infection, as opposed to some other structure within the tissue. When this result is combined with the PMF results, there was clear evidence that the peptides that had been detected mass spectrometrically and mapped spatially were of viral origin. We also note that the difference in correlational values is dependent on both intensity and location. Some peptides naturally ionise with different efficiencies; therefore, we posit that the difference observed in correlation coefficient results from this. 

## 5. Conclusions

We developed a comprehensive pipeline from sample preparation to data interpretation and analysed the data using two discrete analysis methods to describe a robust and reliable pipeline that offers internal validation for MSI data generated in a simple TOF/TOF instrument.

## Figures and Tables

**Figure 1 proteomes-10-00033-f001:**
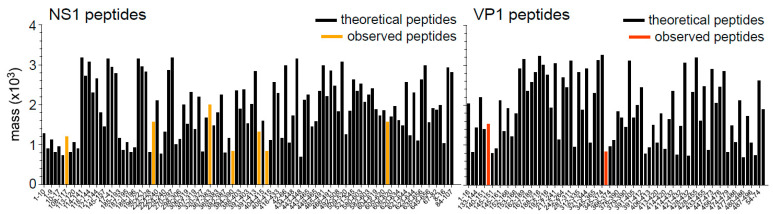
Theoretical tryptic digests of NS-1 and VP-1 demonstrating the number of possible peptides and those detected during MSI. Note the large difference in relative sizes of both proteins with VP1, containing far fewer possible tryptic peptides. Theoretical digests are in Appendix A.

**Figure 2 proteomes-10-00033-f002:**
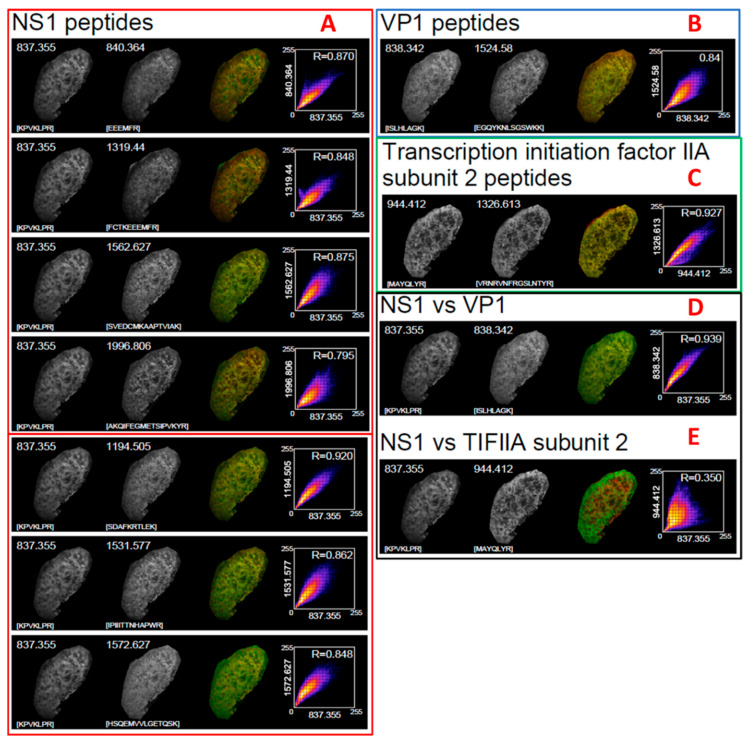
Results of correlational analysis of all detected peptides. (**A**): NS1 peptides, (**B**): VP1 peptides, (**C**): TIIFA2 peptides, (**D**): NS1 compared to VP1 peptides, (**E**): Comparison of NS1 and TIFIIA peptides.

**Table 1 proteomes-10-00033-t001:** Table of all peptides identified by PMF. Masses used in Correlational analysis are in bold.

Viral Protein	Residue Number	Expected Mass [M + H]^+^	Observed Mass [M + H]^+^	Mass Difference (m/z)	Sequence
**NS1**	10–17	958.5904	961.343	−2.7526	RSLSALRR
	**108–117**	**1194.6477**	**1194.505**	**0.1427**	**SDAFKRTLEK**
	**222–236**	**1562.7916**	**1562.627**	**0.1646**	**SVEDCMKAAPTVIAK**
	**367–383**	**1997.0524**	**1996.806**	**0.2464**	**AKQIFEGMETSIPVKYR**
	**384–390**	**837.5668**	**837.355**	**0.2118**	**KPVKLPR**
	**391–403**	**1531.8743**	**1531.577**	**0.2973**	**IPIIITTNHAPWR**
	**404–413**	**1319.5758**	**1319.44**	**0.1358**	**FCTKEEEMFR**
	**408–413**	**840.3556**	**840.364**	**−0.0084**	**EEEMFR**
	449–465	1575.7656	1576.626	−0.8604	GGQACAGGQSAGSLQRK
	**606–619**	**1572.7686**	**1572.627**	**0.1416**	**HSQEMVVLGETQSK**
**VP1**	**140–152**	**1524.7805**	**1524.58**	**0.2005**	**EGQYKNLSGSWKK**
	145–152	919.4996	919.369	0.1306	NLSGSWKK
	**366–373**	**838.5145**	**838.342**	**0.1725**	**ISLHLAGK**
	406–412	780.4111	781.271	−0.8599	YRTGGAR
**TIFIIA**	**1–7**	**944.4585**	**944.412**	**0.0465**	**MAYQLYRN**
	**56–66**	**1325.5927**	**1325.6840**	**−0.0913**	**RVNFRGSLNTYRF**

## Data Availability

The mass spectrometry proteomics data have been deposited to the ProteomeXchange Consortium via the PRIDE partner repository with the dataset identifier PXD034851.

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
