# Peer review of "Viral Biomarker Detection and Validation Using MALDI Mass Spectrometry Imaging (MSI)"

_proteomes, 2022, doi:10.3390/proteomes10030033_

Round 1
Reviewer 1 Report
The manuscript of Proteomes-1840974 is a methodological study on mass spectrometry imaging and mouse kidney parvovirus from Metabolomics and Lipidomics Core Facility, University of Technology Sydney. MSI was performed on serial tissue to test and validate the efficacy of the new MSI workflow.
The ability to effectively analyze formalin-fixed, paraffin-embedded (FFPE) tissue via proteomic techniques offers an invaluable potential for clinical and biomarker research of diseases. Issues of removing crosslinks and accounting for formaldehyde modifications bioinformatically remain to be fully addressed. It would be important to discuss how potential effect of tissue preparation for imaging were excluded in this study, including any effect of tryptic digestion, methylene hydrolysis and α-Cyano-4-hydroxycinnamic acid matrix on MSI and correlational analysis.
After reading through lines 189 – 199, I’d appreciate further clarification of the statement: “This strongly suggests that the co-localised peptides for both NS1 and VP1 did in fact belong to the viral proteome and were not unrelated endogenous mouse associated peptides.”
The same question applies to the text in lines 253 – 258, “The result of this correlational analysis showed clearly that the peptides that were considered to belong to the viral proteins co-occurred with similar intensity with a correlation R2 value of 0.79 and 0.92 for all peptides and did not co-occur with the negative control peptides from TFIIA. When this result is combined with the PMF results there was clear evidence that the peptides that had been detected mass spectrometrically and mapped spatially were of viral origin.”
Lines 173 -175, “Peptide mass fingerprinting (PMF) was then used to screen detected MS1 masses and revealed the detection of 14 peptides from target viral proteins. This included confident identifications of eight peptides from NS-1 and two from VP-1 (Table 1).” Where are the 14 peptides from target viral proteins, eight peptides from NS-1, and two from VP-1 in Fig 1 and Table 1?
There are a number of abbreviations used in the manuscript without definitions, including FFPE, FASTA, MS1, NS1, and VP1.
Reviewer 2 Report
I thank you for the opportunity to review the manuscript titled “Viral biomarker detection and validation using MALDI Mass Spectrometry Imaging (MSI)”, authored by O’Rourke et al. The manuscript is a good study analyzing the scope of MALDI-TOF imaging in identification of viral peptides that is relevant to the readership of the Journal. The study effectively highlights the use of MALDI-MSI for viral peptide detection and its applicability in the clinical situations. However, I have a few concerns regarding the manuscript, which could be addressed by the authors.
1. The authors utilize samples from a previous study where they have identified a specific virus infecting mouse kidneys to perform the analysis and the study is referred to in the manuscript. However, there is not enough information on the samples and the nature of virus in the introduction. I feel that the readers would benefit from a brief background within this manuscript on the virus they are investigating to provide context to the study. The authors provide this information in the discussion section of the manuscript (lines 204-212) but this might be more relevant in the introduction.
2. There is extensive reference to the authors previous works including some critical methodologies that inhibits the reader from understanding critical aspects. Especially since many of these articles are unavailable for access. For example, in the lines 152-154, the authors describe an essential process of inspecting images to distinguish genuine spectra from that of the artifact which is a major concern in such studies. However, the readers could benefit from a concise version of how this is carried out instead of the reference alone. Also, one of the two references provided was not accessible at all.
Though I appreciate the hard work that has gone into these previous studies, absence of critical information and mere referencing severely impairs the reading and comprehension of the manuscript.
3. In my understanding, the authors have attempted to provide the number of theoretical peptides generated and the ones detected by the experiment in Figure 1. However, the interpretation is not very straight forward or easily interpretable. I request the authors to please provide the protein sequence (or at least the protein identifiers, if available) for the two viral proteins that was used for the generation of the theoretical peptides. Perhaps, they could mention the total number of theoretical peptides generated from each protein and/or indicate the peptides detected within the sequence in addition to the current representation to guide the readers.
For example, in the figure legend (line 182), the authors refer to the variation in the protein sizes that is not evident from the graph which shows the residues in the x-axis (axis is also not labeled).
4. In line 174, the authors mention that they identified a total of 14 viral peptides by PMF of which 8 and two were confident identifications respectively. Could the authors please comment on the remaining 4 peptides and the basis of the confidence determination here?
5. The authors have identified a significant number of viral peptides in the study, however, I find that the manuscript would benefit from a sample spectra showing the intensity of at least one peptide for each of the viral and the control proteins.
6. The image correlation analysis shows considerable correlation within the viral peptides and that of the peptides from the control protein. However, I notice that the correlation of the detected peptides from NS1 protein shown in the figure 2 is always with that of the “KPVKLPR” peptide. I was curious to know as to how the correlation fared between the other peptides (say between “SVEDCMKAAPTVIAK” and “FCTKEEEMFR”). The correlation values between each of these peptides could be perhaps presented in the form of a matrix either in the main manuscript or in the supplementary files.
Also, I was curious as to know what the author thinks could be a possible reason for the relatively lower correlation between some of the peptides of the NS1 protein. Perhaps a comment in the discussion would benefit the readers.
Minor points:
1. In the abstract, the authors mention a correlation value of >0.81, where as the lowest correlation value in the results shown in figure 2 is that of 0.79. Is 0.81 considered a threshold here?
2. The manuscript could benefit from a spell-check. For example, Line 30: spelling of “simple MALDI TOF/TOF”.
There were multiple instances where it was not clear what the author intended to say, especially in the discussion section. For example, in line 219, the authors state “Here we utilise a hybrid identification workflow to achieve reliable results so long as inbuilt validation is integrated at each stage of the workflow.” It was not clear to me what the author intended to say. Could you please clarify what this statement means?
Similarly, lines 227-229; 238-240; 243-245 could benefit from a more direct and simplified language in conveying the author's intention.
Reviewer 3 Report
The authors developed a novel approach by integrating visual image analysis and bioinformatics to confirm and validate peptides which are identified in the TOF/TOF MSI experiment. Moreover, they applied the current method to characterize the progression of viral infection in mouse kidney. Although this is an interesting study, there are few major and minor issues that need to be addressed
1. The correlation analysis between TFIIA2 and viral NSI showed an R2 of only 0.32. What is the correlation between TFIIA2 and VP1? Does it show a similar number? It would be better to have that number in the manuscript. The reviewer would like to see that part in Fig 2 or at least in the supplement section.
2. The authors aimed to develop a hybrid method that combines bioinformatics and image-based methods for the identification of biomarkers. However, the current study missed the comparative study between two biological duplicates. For instance, if few peptides are identified in one sample, will it be possible to detect the same peptide group in another biological duplicate experiment by following the current approach? If yes, it is strongly recommended to keep a quantitative study between the biological duplicates.
3. The author claimed to characterize the progression of viral infection in mouse kidney. However, the result section only showed that NS1 and VP1 belong to the viral proteome, a validation of their method with a single experiment. How does a single experiment characterize the progression of viral infection?
4. Including the potential limitation and explaining the implication of the limitation always enriches the reader’s understanding and supports future investigation. Therefore, the reviewer would like to see the limitations of this approach in the discussion section.
5. Notably, self-citation by the authors reached over 40 % in the current manuscript and therefore, the reviewer suggests the authors to update the references if applicable.
.
Reviewer 4 Report
The paper “Viral biomarker detection and validation using MALDI Mass Spectrometry Imaging (MSI) by O’Rourk et al. Looks like a quite interesting study. The paper deals with a relevant topic in medical field.
The MALDI TOF-type mass spectrometer allows to identify with certainty and precision types and strains of microorganism responsible for a certain pathology by simply taking a sample of body fluids from the patient and subjecting it to spectrometric analysis even without prior treatment. The use of MALDI TOF technology mainly rely to clinical practice for the identification of pathogenic microorganisms responsible for the most common infections, the mass spectrometer has landed in hospitals in many countries, becoming part of the usual and common diagnostic analytical techniques alongside more traditional assays, spectrometer analysis in fact has numerous advantages, including the speed with which it allows sure results to be obtained. The only limitation is the availability, in the international databases of health care providers, of the spectrum corresponding to the one investigated that one wants to compare. However, since the MALDI TOF method, already in use for for a long time in all the world's scientific research laboratories for the characterization precisely of molecular structures, has been recently exported to the medical field the creation of entire libraries of molecular spectra databases have sprung up and are springing up. Moreover, the most modern MALDI TOF tools allow one to create one's own personal library and share it online with other health care professionals, hospital companies and research centers. In addition, as authors reported rather than having a dedicated instrument for MSI, ion sources and attachments can be purchased that modify existing QTOF or Orbitrap instruments.
Minor revision
Despite some limitations, the proposed approach of a hybrid bioinformatic/image-based method for the identification and validation of viral biomarkers to contribute to research activity of teams who’re still using simple TOF/TOF instruments may be useful.
However, Authors are asked to address the transferability of this approach to clinic, being the MALDI TOF/TOF mainly used in Hospitals by Physicians
Line 30 simle has to be corrected in simple and
Line 30 TO has to be corrected in TOF
Line 175 Authors refer to eight peptides while either in Figure 1 and Figure 2 A seven peptides are reported for NS1 protein.
Figure 1 caption sizs has to be corrected in sizes
Round 2
Reviewer 1 Report
-
Reviewer 3 Report
The Authors worked on the reviewers' comments and made significant improvement! I believe this manuscript is now of a publishable standard.
Reviewer 4 Report
The authors have made efforts to improve the quality of the article. The article can now be accepted for publication in Proteomes.